# YOLOv8n-BWG-enhanced drone smoke detection: Advancing environmental monitoring efficiency

Yaojun Zhang, Guiling Wu[ID]*

School of Information Engineering, Xinyang Agriculture and Forestry University, Xinyang, China

* guiling@xyafu.edu.cn

**Data availability statement:** All relevant data for this study are publicly available from the figshare repository (https://doi.org/10.6084/m9.figshare.28512971).

## Abstract

The precise monitoring and localization of industrial exhaust smoke emissions play a crucial role in environmental management. Existing methods encounter challenges like intricate detection environments, small-scale targets, and extensive model parameters. This study presents an advanced drone smoke detection model, YOLOv8n-BWG, building on YOLOv8. It introduces a novel BC2f structure into the backbone network, leveraging an adaptive query mechanism to minimize computational and storage demands while boosting feature extraction efficiency. Additionally, the study employs a dynamic sample allocation strategy to refine the loss function, enhancing the model's sensitivity to small targets. It also integrates a lightweight convolution, GSConv, in place of traditional convolution techniques. GSConv employs a channel grouping approach, streamlining model parameters without sacrificing accuracy. Results on a specialized dataset reveal that YOLOv8n-BWG outperforms YOLOv8n by increasing the mean Average Precision (mAP) by 4.2%, boosting recognition speed by 21.3% per second, and decreasing both the number of floating-point operations (FLOPs) by 28.9% and model size by 26.3%. Significantly, deploying YOLOv8n-BWG on drones yielded promising outcomes in smoke detection, offering innovative approaches and insights for effective smoke monitoring practices.

## Introduction

Effective smoke monitoring can significantly alleviate the negative impacts of industrial emissions on both the environment and human health, thereby enhancing air quality and facilitating sustainable development [1]. As of 2022, fossil fuels account for over 85% of China's energy consumption. The combustion of these fuels emits substantial pollutants, leading to severe environmental challenges and considerable public health risks [2]. Consequently, it is crucial to monitor and accurately pinpoint smoke emissions from chimneys.

Early research on monitoring chimney smoke emissions primarily utilized sensors placed within the monitoring area alongside established air quality monitoring stations. These stations transmitted data back to a base station (BS) for detailed inspection and analysis to pinpoint pollution sources [3]. However, this approach encountered several issues,

**Funding:** This study was financially supported by the Key Scientific Research Project of Higher Education Institutions in Henan Province in the form of a grant (24B520035). This study was also financially supported by the Scientific and Technological Breakthrough Project of Henan Province in the form of a grant (252102111173). The funders had no role in study design, data collection and analysis, decision to publish, or preparation of the manuscript.

**Competing interests:** There is no potential conflict of interest in our article, and all authors have seen the manuscript and approved it to submit to your journal.

including limited flexibility, restricted monitoring range, high costs, and susceptibility to environmental interference. With the progression of science and technology, innovative visual monitoring strategies that integrate Unmanned Aerial Vehicles (UAVs) with target detection algorithms have gained prominence [4]. This novel approach employs target detection algorithms to monitor smoke emissions and utilizes UAVs for surveillance, offering a highly efficient and precise method for chimney smoke emission monitoring. It effectively addresses the drawbacks of traditional methods, providing a more practical and superior solution for monitoring smoke emissions [5].

Target detection algorithms are broadly classified into two types: traditional and deep learning-based approaches. Traditional methods, such as edge detection [6], Histogram of Oriented Gradients (HOG) [7], and Support Vector Machine (SVM) [8], depend significantly on manual feature extraction, leading to limited generalization capabilities. In contrast, deep learning-based algorithms offer enhanced generalization through automated feature extraction. These algorithms are further categorized into single-stage (e.g., SSD [9], YOLO series [10]) and two-stage (e.g., R-CNN [11], Fast R-CNN [12], Faster R-CNN [13]) approaches. Two-stage algorithms distinguish themselves by first generating region proposals and then conducting object detection within these regions, effectively separating the detection process into two distinct phases [14]. For example, Yuan et al. [15] demonstrated the application of Faster R-CNN in monitoring industrial structures in high-resolution satellite images, showcasing its ability to accurately detect targets under diverse imaging conditions. Similarly, Han et al. [16] utilized Faster R-CNN for chimney detection in urban imagery from Google Maps, achieving precise results. Despite the success of two-stage algorithms in achieving high accuracy, their extensive model parameters and slower detection speeds pose deployment challenges in systems with limited resources, such as unmanned aerial vehicles (UAVs).

To optimize real-time target detection and enhance compatibility with edge devices, single-stage algorithms are generally favored. These methods approach object detection as a regression challenge, leveraging convolutional neural networks to directly predict bounding box locations and categories. This eliminates the need for candidate region generation and classification steps required in two-stage algorithms, thereby significantly improving detection speed. Such efficiency renders single-stage algorithms ideal for real-time applications and deployment in resource-constrained environments [17].

In specific studies, Cao et al. [18] modified the SSD framework by replacing its backbone with MobileNets, making it more suitable for mobile and embedded applications. However, this modification compromised detection accuracy, rendering it insufficient for smoke detection requirements. Zhao et al. [19] proposed an enhanced Fire-YOLO algorithm that expands feature extraction capabilities across three dimensions. While this improved small smoke target detection, its processing speed remained inadequate for real-time applications. Wang et al. [20] developed Light-YOLOv4, a lightweight model substituting CSPDarknet53 with MobileNetv3, but achieved suboptimal mean average precision (mAP) for smoke detection standards. Geng et al. [21] introduced an improved YOLOv5-based UAV fire smoke detection system utilizing focalNextBlock modules from CFNET to enhance multi-scale information integration and reduce parameters. Nevertheless, this method underperformed in detecting small smoke targets. Yang et al. [22] enhanced YOLOv5 by integrating PConv for drone-based forest fire detection, reducing model parameters and accelerating processing speed, though with compromised detection accuracy. Xu et al. [23] combined YOLOv5 and EfficientDet with an additional verification classifier, achieving higher smoke detection accuracy but resulting in excessive model size and slow inference speed, making it unsuitable for UAV deployment.

To address high computational complexity in UAV detection models, Sun et al. [24] developed a lightweight YOLOv7-based smoke detection model. While significantly reducing computational costs compared to original YOLOv7, its detection accuracy remained insufficient. Luan et al. [25] proposed YOLO-CSQ, an improved YOLOv8-based UAV fire detection method featuring a Quadrupled-ASFF detection head for weighted feature fusion. Although this enhanced multi-scale detection robustness, the model struggled to distinguish between clouds and smoke, leading to frequent false positives.

In summary, significant advancements have been achieved in the field of smoke detection through prior research. Nevertheless, existing methodologies encounter challenges, including false positives amidst complex backgrounds, notably with clouds and smoke; the overlooking of small-scale smoke; and the constraints imposed by substantial model parameters on deployment within unmanned aerial vehicle (UAV) systems. Addressing these concerns, this paper presents enhancements utilizing the YOLOv8n model. The key contributions are outlined as follows:

1) The development of an innovative BC2f structure incorporates an adaptive query mechanism, diminishing computational and storage demands. This enhancement not only augments the model's input content sensitivity but, by embedding the BC2f structure within the backbone network, also sharpens focus on smoke detection. It effectively suppresses irrelevant features and noise, thereby mitigating false positives associated with cloud and smoke misidentifications.

2) Introduction of a dynamic sample allocation technique alongside loss function re-optimization, significantly bolstering the model's focus on smaller instances. This adjustment notably decreases the likelihood of overlooking minute smoke traces.

3) The substitution of standard convolution with a novel lightweight convolution, GSConv, which organizes input channels into groups to perform independent depthwise separable convolutions, followed by a channel mixing process. This approach efficiently trims model parameters while preserving accuracy.

4) Deployment of the enhanced YOLOv8n model in UAV systems, equipped with a binocular stereo vision system for calculating three-dimensional smoke location data. Accurate positions are determined via established conversion equations, offering more detailed and dependable environmental sensing information.

## Smoke detection model based on improved YOLOv8

### YOLOv8 network structure

Introduced in January 2023 by Ultralytics, YOLOv8 [26] is an advanced single-stage object detection algorithm. This model surpasses its predecessor, YOLOv5, in detection speed and accuracy, making it ideal for deployment on constrained edge devices that require high inference speed. YOLOv8 encompasses five variations: YOLOv8n, YOLOv8s, YOLOv8m, YOLOv8l, and YOLOv8x, which differ in size and computational complexity but share a common underlying principle. YOLOv8n, the smallest and least computationally demanding variant, is particularly well-suited for edge devices. This study focuses on the YOLOv8n model, which is structured into four main components: the input end, backbone network, neck network, and detection end, as depicted in Fig 1.

### Embedding BiFormer attention mechanism

Unmanned Aerial Vehicles (UAVs) capture images with complex backgrounds, necessitating detection models to prioritize key information over background noise. Attention

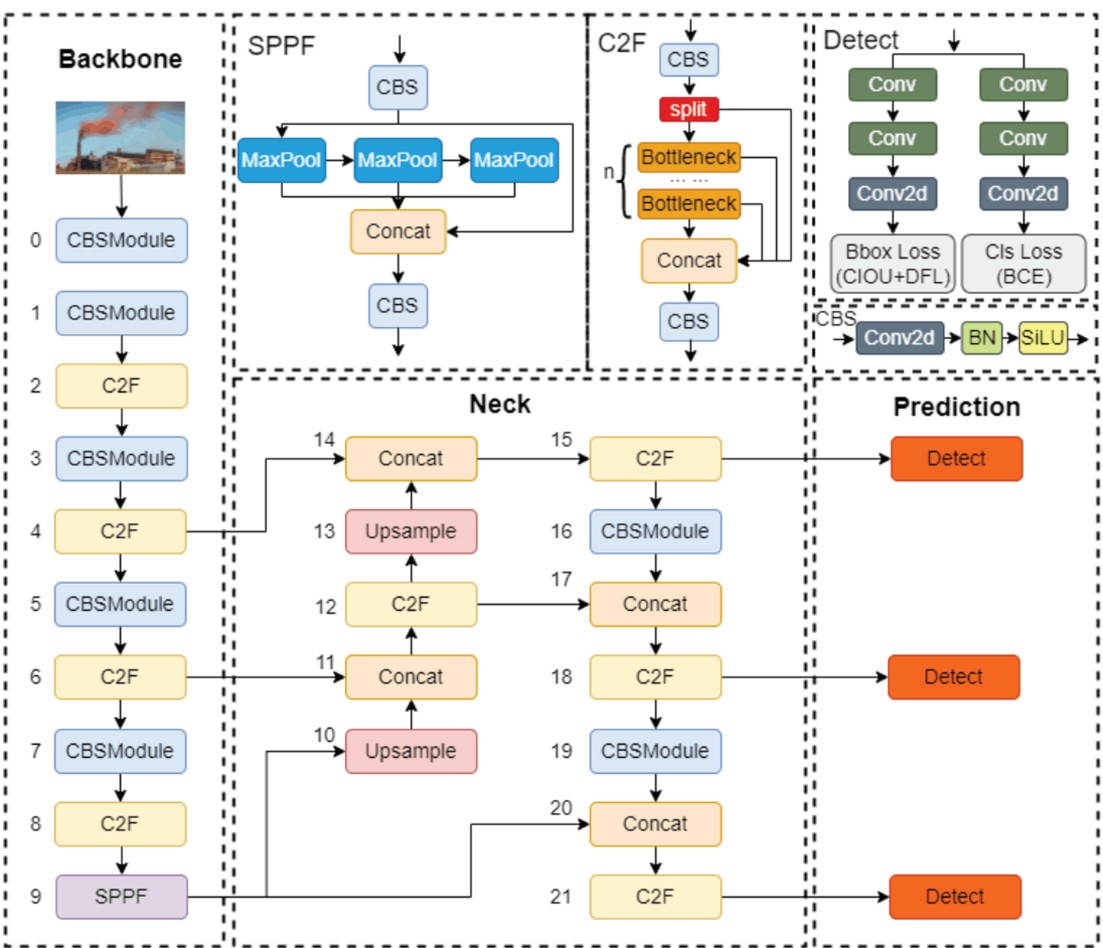

**Fig 1. YOLOv8n structure diagram.**

mechanisms, such as those proposed in references [27], effectively enhance focus on relevant targets like smoke, while minimizing distractions from ineffective features. However, widely recognized attention frameworks including SENet [28], CBAM [29], and GAM [30], encounter limitations related to computational intensity and memory demands, which pose deployment challenges on UAV platforms. Addressing these limitations, Zhu et al. [31] introduced the BiFormer attention mechanism, characterized by its efficient resource use through the adoption of sparse queries targeting specific key-value pairs, reducing the computational and memory footprint.

Incorporating the BiFormer mechanism into the YOLOv8n model, this study presents a novel BC2f structure. This structure employs BiFormer attention at both the input and output stages, focusing on sparse queries to enhance model performance on smoke detection. The BC2f aims to reconcile the disparity between lower feature maps, which provide detailed locational insights but lack contextual information, and higher feature maps, which offer rich semantic context but are deficient in locational details. In response to the limitations of Fig 2(a), by integrating BiFormer attention, the proposed structure seeks to optimize the balance between these feature maps, thereby improving smoke detection accuracy and efficiency, as illustrated in Fig 2(b).

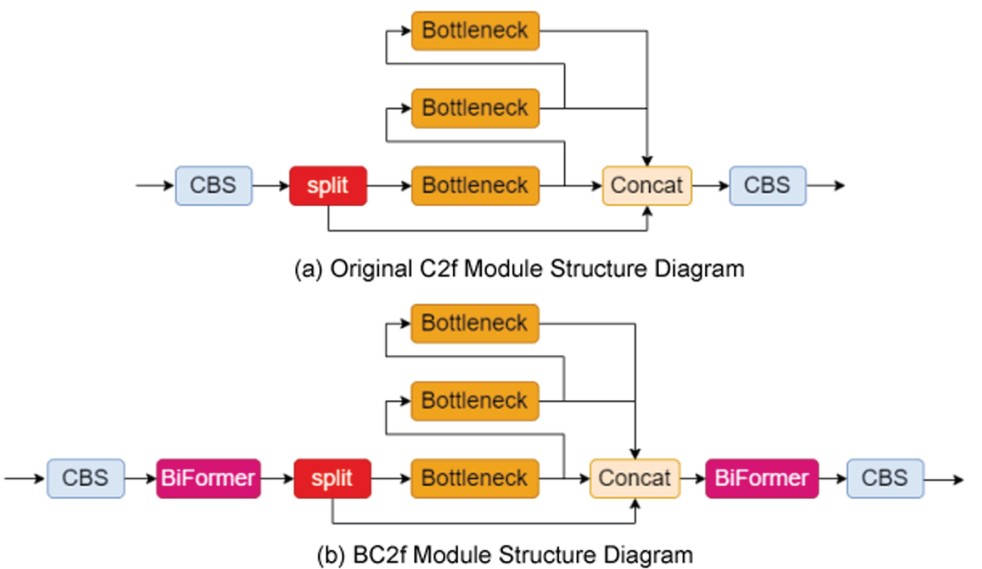

**Fig 2. C2f and BC2f module structure diagrams.**

The BiFormer attention mechanism's design, detailed in Fig 3, begins with adaptive querying to eliminate irrelevant key-value pairs from coarse input feature areas, followed by attention processing. This approach substantially lowers computational and storage demands, augmenting the model's input sensitivity. Initially, BiFormer utilizes overlapping block embedding to refine input features. Subsequently, it employs block merging modules from the second layer onwards to reduce spatial resolution and augment channel count, followed by feature enhancement via multiple BiFormer blocks. Each block starts with a 3x3 depthwise convolution for encoding relative positional cues, succeeded by a BRA module and a

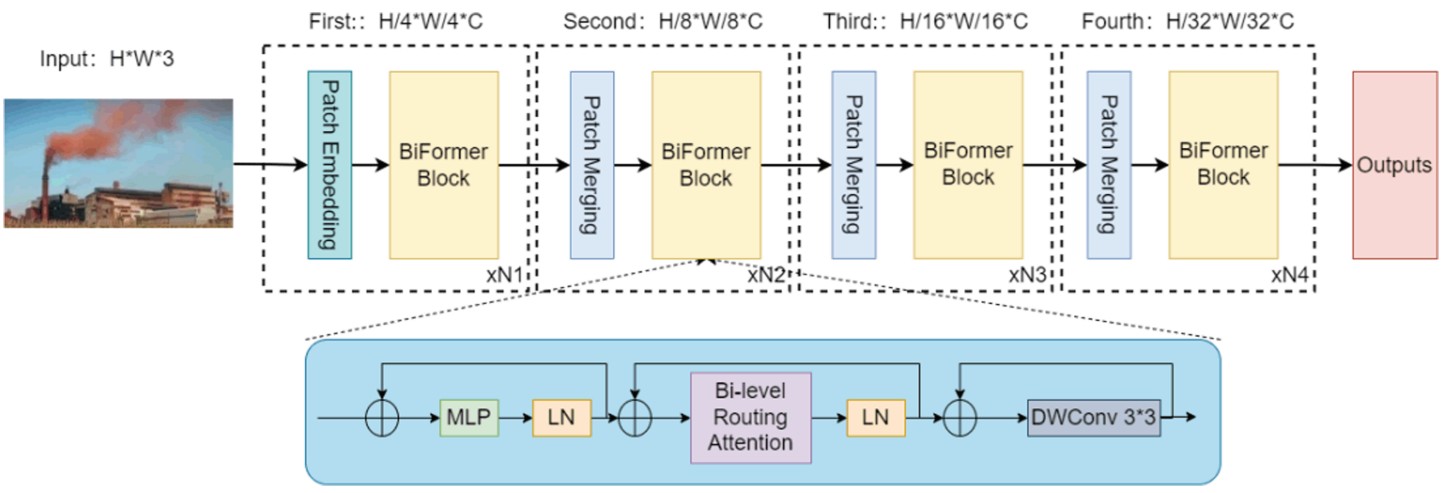

**Fig 3. BiFormer structure diagram.**

dual-layer multi-layer perceptron (MLP) module with an expansion rate of e, to capture and embed cross-positional relationships effectively.

To more clearly compare the differences between including and not including the BiFormer attention mechanism, thermal maps of the last layer of the backbone network were printed, as shown in Fig 4. It is distinctly observed that the incorporation of the BiFormer attention mechanism enhances the backbone network's focus on chimney smoke, effectively suppressing ineffective features and noise.

## WIoU loss function

In deep learning networks, the impact of loss functions on network performance is increasingly emphasized. Traditional loss functions primarily focus on the overlap between predicted and actual bounding boxes, leading to a meaningless function when there is no intersection between them [32]. In such cases, the loss function fails to effectively indicate the distance between the predicted and actual boxes, thereby hindering further optimization of the network model. This constraint proves ineffective for the recognition of small targets [33].

In the YOLOv8n network model, the loss function employs Distribution Focal Loss and CIoU loss function. The CIoU loss function, building upon the DIoU loss function, introduces bounding box scale loss and aspect ratio loss. These enhancements aid in improving the accuracy of regression predictions. However, the CIoU loss function does not account for mismatches between the orientation of actual bounding boxes and predicted ones, leading to slower network convergence and other negative impacts. Therefore, this paper improves the YOLOv8n model by optimizing the loss function with Wise-IoU (WIoU) [34].

WIoU is a strategy designed to optimize bounding box and classification losses. When processing training datasets containing smaller targets, WIoU considers geometric factors such as distance and aspect ratio to increase the weight proportion of small targets, thereby intensifying focus on small samples. WIoU introduces a distance attention mechanism, using two layers of attention mechanisms to prevent slow convergence and reduce harmful gradients caused by small sample data, thus enhancing the model's generalization ability. RW-IoU represents the loss of high-quality anchor boxes. The WIoU formula is as follows:

$$L_{WIoU} = R_{WIoU}L_{IoU} \tag{1}$$

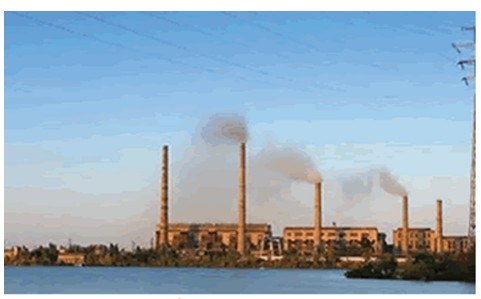 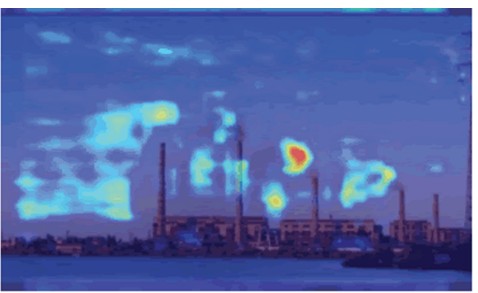 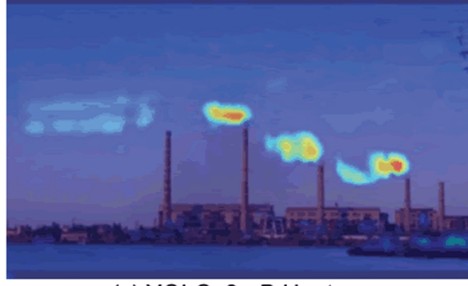

(a) Original Image (b) YOLOv8n Heatmap (c) YOLOv8n-B Heatmap

**Fig 4. Heatmap changes before and after adding BiFormer attention mechanism.**

$$R_{WIoU} = \exp\left(\frac{(x - x_{gt})^2 + (y - y_{gt})^2}{(W_g^2 + H_g^2)^*}\right) \tag{2}$$

In the context of the study, $L_{IoU}$ represents the degree of overlap between the predicted box and the actual box; $w_g$ is the width of the smallest enclosing box; $H_g$ is the height of the smallest enclosing box; $x_{gt}$ is the horizontal coordinate of the center point of the actual box; $y_{gt}$ is the vertical coordinate of the center point of the actual rectangular box.

On this basis, to prevent significant harmful gradients from small samples, a smaller gradient gain was introduced, focusing the bounding box regression on anchor boxes of normal quality. WIoUv2 and WIoUv3, based on the construction of bounding box loss, incorporate monotonic and non-monotonic focusing mechanisms, respectively. WIoUv2 constructs a monotonic focusing coefficient $L_{IOU}^{\gamma^*}$, as follows:

$$L_{WIOUv2} = L_{IOU}^{\gamma^*} L_{WIOUv1}, \gamma > 0 \tag{3}$$

Where $L_{IOU}^{\gamma^*}$ is the gradient gain, which decreases with the decrease in overlap.

WIoUv3, by defining an outlier b, constructs a non-monotonic focusing mechanism, building a non-monotonic focusing coefficient applied to WIoUv1, as shown below:

$$L_{WIOUv3} = rL_{WIOUv1} \tag{4}$$

$$r = \frac{\beta}{\delta\alpha^{\beta-\delta}} \tag{5}$$

Where $r$ is the non-monotonic focusing coefficient; $\alpha$ and $\gamma$ are hyperparameters.

WIoU employs a dynamic non-monotonic focusing mechanism to assess the quality of anchor boxes and utilizes gradient enhancement to construct an attention-based bounding box loss. During the experiment, the study conducted ablation experiments on the loss function by comparing WIoUv1, WIoUv2, and WIoUv3 to find the best loss function for the dataset.

## GSConv module

With advancements in network accuracy, there is a tendency for an increase in the number of network layers, leading to a notable decrease in detection speed. The enhanced YOLOv8 model, while benefiting from additional modules, faces a consequential rise in parameters and computational burden. This escalation complicates deployment on unmanned aerial vehicle (UAV) platforms, necessitating a balance between parameter volume and accuracy. To address this, our study implements a lightweight convolution technique to refine the model.

This study introduces the GSConv [35] lightweight convolution method as a replacement for standard convolutions, effectively diminishing the model's parameter count without sacrificing detection accuracy. GSConv aids YOLOv8 in lowering computational requirements while preserving robust performance, thus improving model feature extraction efficiency.

GSConv, rooted in depthwise separable convolution (DSC), standard convolution (SC), and channel mixing operations, aims to enhance information transfer and collaborative

learning by weight sharing among groups. This strategy boosts local feature extraction and minimizes parameter count, enhancing network trainability and generalizability. GSConv's architecture, detailed in Fig 5, showcases its innovative approach.

When the convolution size is $f \times g$, $x \times y$ is the output size, $M$ is the number of input channels, and $N$ is the number of output channels, the computation of standard convolution $SC$ is as shown in Eq (6).

$$SC = f \times g \times x \times y \times M \times N \tag{6}$$

The calculation for Depthwise Separable Convolution (DSC) is as shown in Eq (7).

$$DSC = f \times g \times x \times y \times M + x \times y \times M \times N \tag{7}$$

Therefore, the compression ratio between DSC and SC is as shown in Eq (8).

$$\frac{DSC}{SC} = \frac{f \times g \times x \times y \times M + x \times y \times M \times N}{f \times g \times x \times y \times M \times N} = \frac{1}{N} + \frac{1}{f \times g} \tag{8}$$

In the GSConv lightweight convolution, each convolution kernel consists of a main convolution kernel and a secondary convolution kernel. The main convolution kernel, serving as the core, is responsible for the primary computation, while the secondary convolution kernel supplements the computation of the main kernel, thus enabling grouped convolution and significantly reducing computational load. Its computation formula is as shown in Eq (9).

$$(f^{*}g)(n) = \sum_{i=1}^{n-1} f(i)g(n \oplus i \oplus r(i)) \tag{9}$$

Where $\oplus$ represents the dissimilar operation, $n$ represents the length of the input, $r(i)$ represents the $i$ th element of the input, corresponding to a random shift amount in convolution. GSConv has a lower time complexity because it retains channel-dense convolution, as shown in Eq (10).

$$time_{GSConv} \sim O\left[ x * y * f * g * \frac{C_2}{2}(C_1 + 1) \right] \tag{10}$$

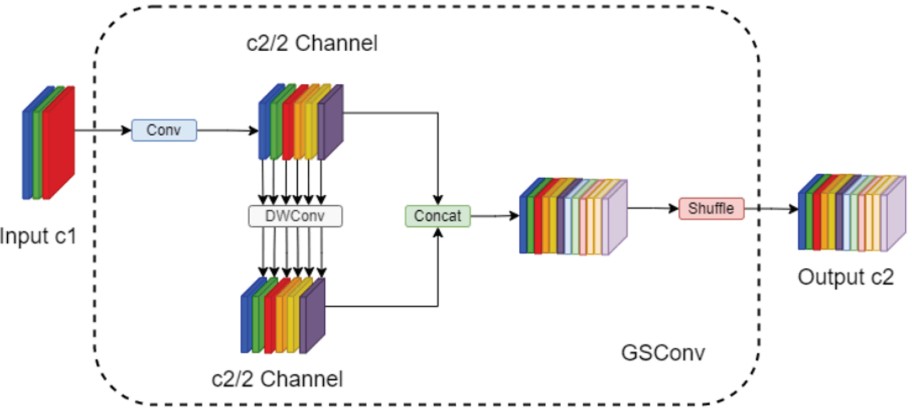

**Fig 5. GSConv module structure diagram.**

The time complexity formulas for Standard Convolution (SC) and Depthwise Separable Convolution (DSC) are as shown in Eqs (11) and (12).

$$time_{SC} \sim O\left(x * y * f * g * C_1 * C_2\right) \tag{11}$$

$$time_{DSC} \sim O\left(x * y * f * g * 1 * C_2\right) \tag{12}$$

The output channel number is C2, and the convolutional kernel channel number is C1. In this convolution operation, by sliding the main and secondary convolutional kernels in horizontal and vertical directions, and employing channel mixing and shuffling strategies, input channels are grouped, and cross-computation is performed within each group. This approach not only enhances network performance but also reduces the model's parameter count.

### The improved YOLOv8n-BWG model

The enhanced YOLOv8n model architecture targets improvements across the backbone network, neck network, and loss function. Initially, it integrates the BC2f module, which employs the BiFormer attention mechanism. This mechanism, through an adaptive querying approach, decreases both computational and storage demands, concurrently improving input content perception. Additionally, the backbone network's feature extraction capabilities are enhanced by substituting its original C2f module with the BC2f structure [36]. The paper also introduces an upgrade to the CIoU loss function by incorporating the WIoU2 function. This enhancement, featuring a dynamic sample allocation strategy, notably increases the model's precision in small sample detection. Moreover, the standard convolution is replaced by the lightweight GSConv convolution. GSConv groups input channels, conducts depthwise separable convolutions independently for each group, and then merges them via channel mixing, thereby reducing model parameters while preserving accuracy. The resultant YOLOv8n model, embodying the BiFormer attention mechanism, WIoU loss function, and GSConv module, is denoted as the YOLOv8n-BiFormer-WIoU-GSConv network (YOLOv8n-BWG model). Fig 6 illustrates its structure, with blue highlighting the BC2f module integration and red indicating the GSConv's implementation.

## Experimental and results analysis

### Dataset establishment

The increasing demand for data on smoke and industrial chimney conditions underscores the importance of such information in supporting research and applications across relevant fields. Yet, a critical issue persists: the scarcity of accessible datasets for utilization. The acquisition of a comprehensive and diverse dataset for model training and testing presents a formidable challenge within the current research and application landscape. To overcome this, our study successfully compiled a dataset comprising 2000 images and 200 videos, sourced through web scraping with keywords including 'smoke' and 'industrial chimney.' Subsequently, these videos were converted into 6100 images via frame extraction. Through a meticulous data cleansing process, which entailed the exclusion of images featuring non-chimney smoke and those of poor quality, a refined collection of 4123 chimney smoke images was curated.The entire data collection and processing procedure strictly adheres to the terms and conditions of each data source, ensuring compliance with public domain or fair use guidelines, thereby guaranteeing the legality and ethical standards of the data collection.Some of the pictures are shown in Fig 7.

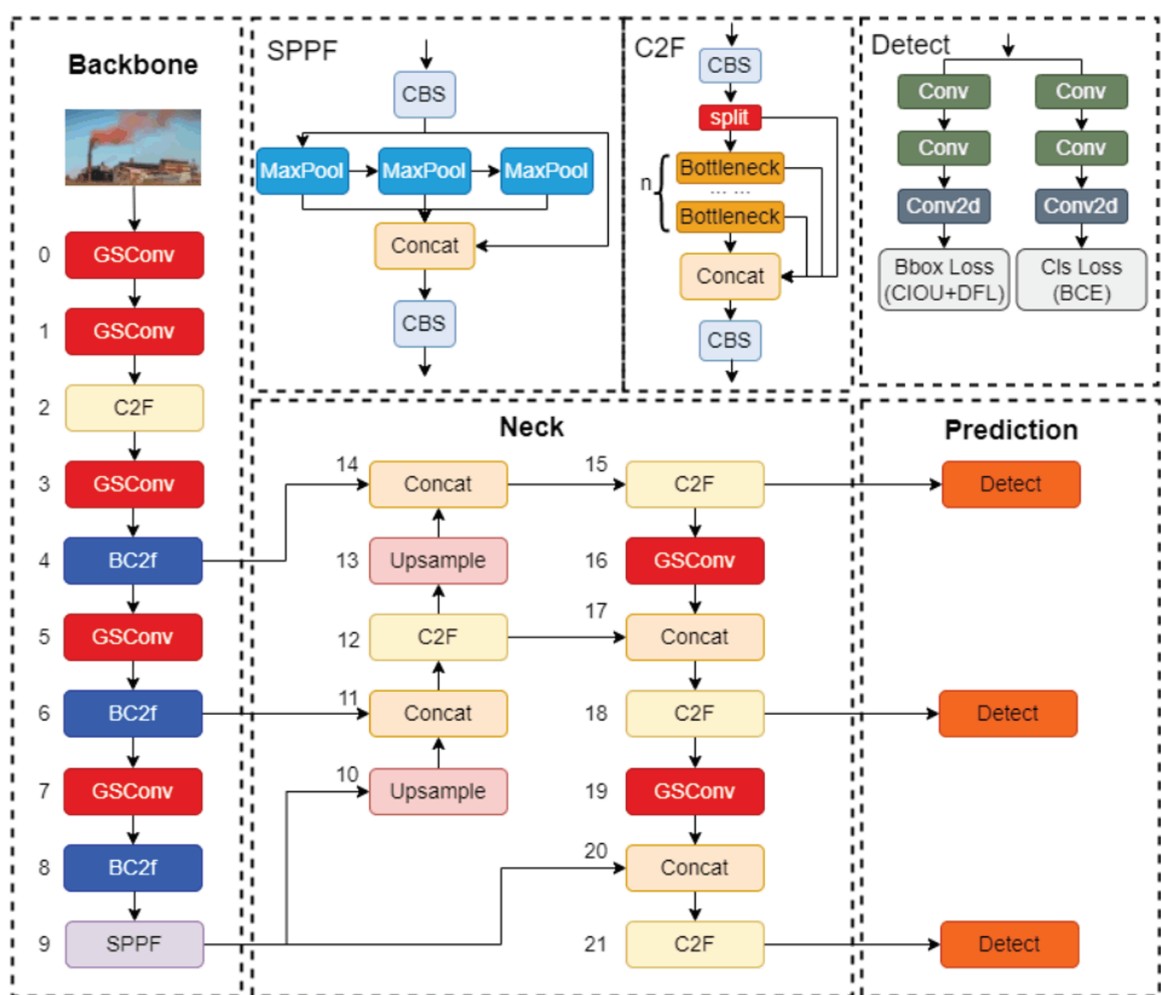

**Fig 6. YOLOv8n-BWG model structure diagram.**

This dataset's assembly is pivotal, providing vital resources for model training and testing, thereby enabling further research and facilitating in-depth investigation into smoke-related issues. The images were annotated using labelimg software, generating XML files in the Pascal VOC format. These files contain crucial information on target categories and locations, essential for network training.

The dataset's limited diversity poses a challenge, as training models could potentially misinterpret noise or extraneous features as relevant characteristics. To mitigate this, the study integrates online data augmentation techniques, such as the introduction of random noise, brightness adjustments, and scaling. These methods produce varied, albeit similar, data in each training cycle, thereby enhancing the model's learning of smoke attributes, its generalization capabilities, and overall accuracy. The augmented dataset was divided into training, validation, and test sets in a 7:2:1 ratio. The augmented images are shown in Fig 8.

Visualization tools facilitated the examination of the dataset, as depicted in Fig 9. Fig 9(a) outlines the dataset's category distribution; Fig 9(b) shows the annotation boxes; Fig 9(c) presents the distribution of target box positions, with axes indicating their center; and

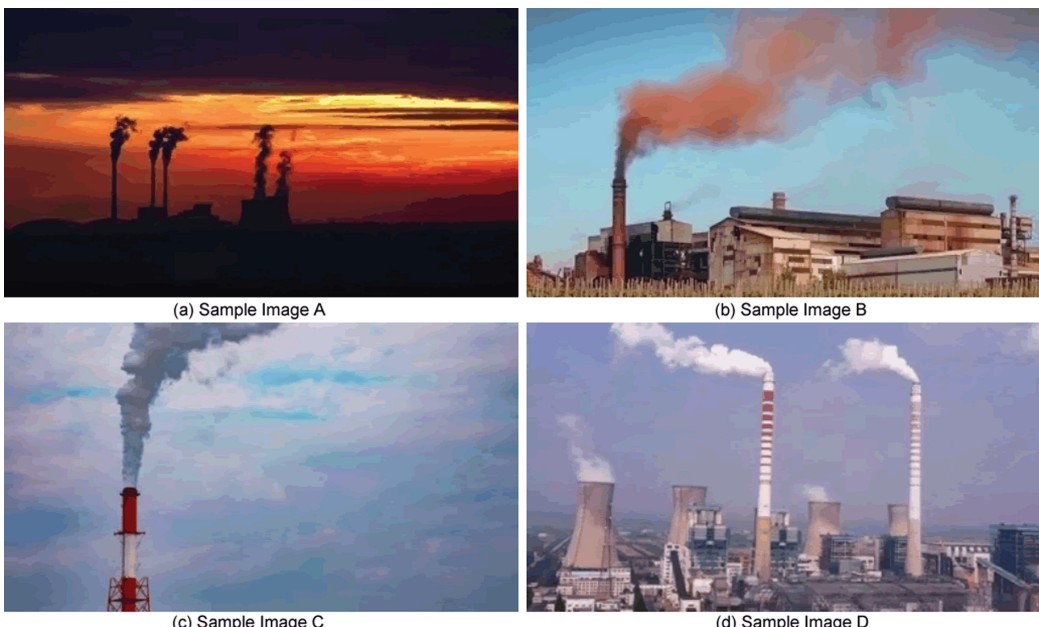

**Fig 7. Dataset sample images.**

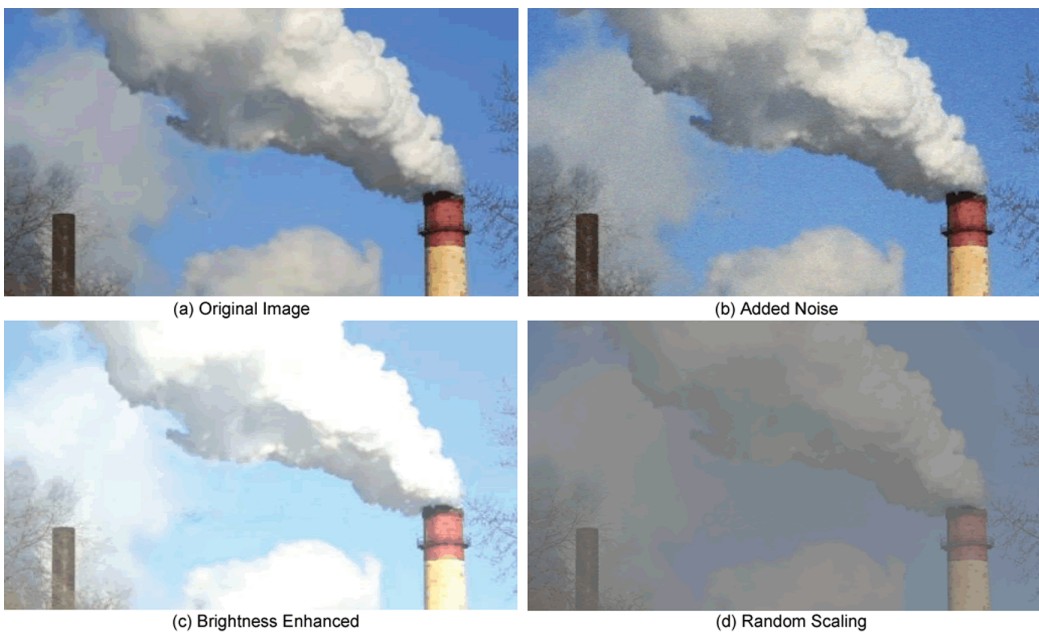

**Fig 8. Enhanced dataset sample images.**

Fig 9(d) details the size distribution of target boxes, with axes for annotated width and height.

Analysis of Fig 9(b) indicates a consistent and uniform size of target boxes. Fig 9(c) demonstrates a uniform distribution of target box centers, and Fig 9(d) shows that most target boxes are small, suggesting the dataset includes numerous small targets.

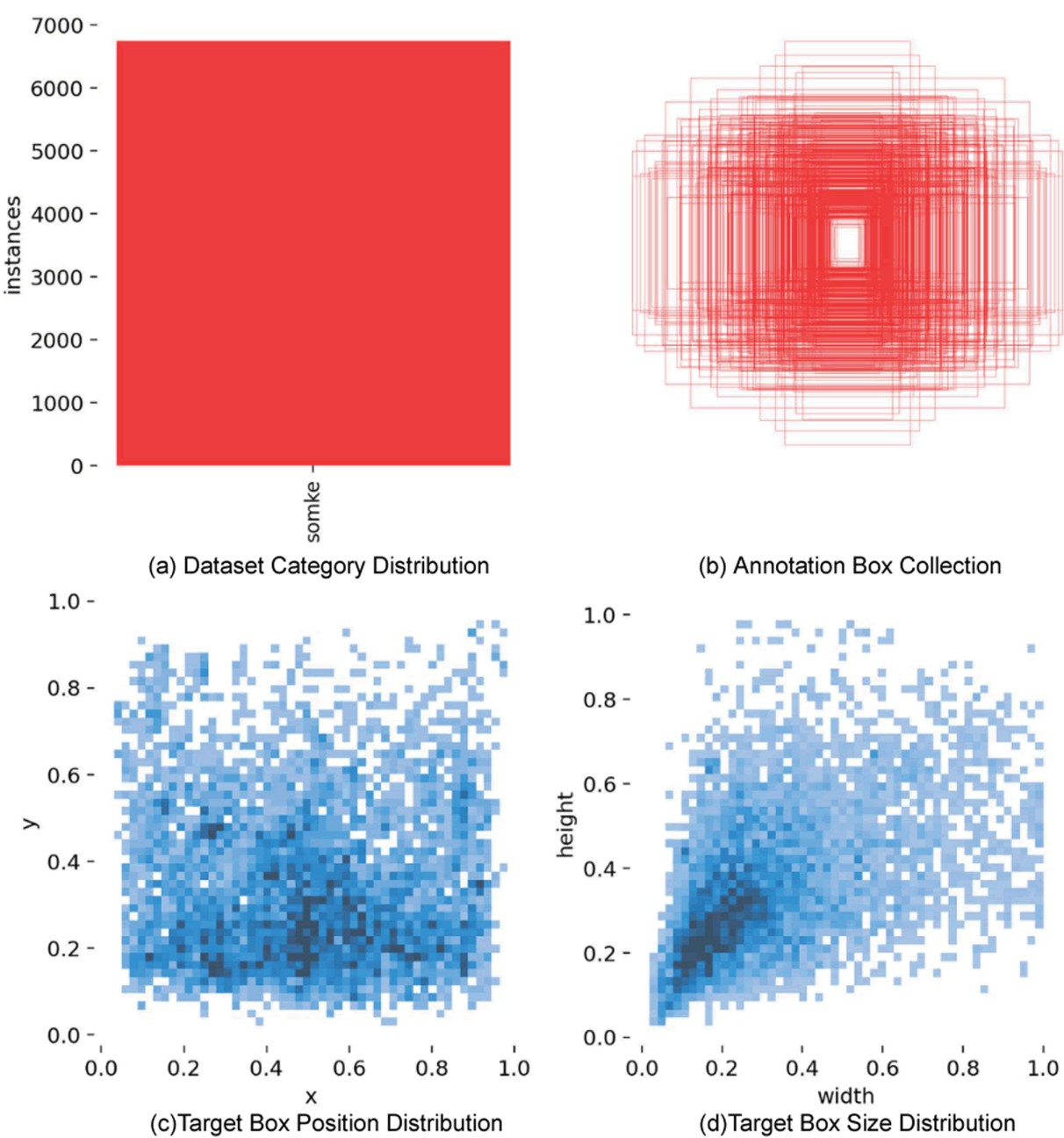

**Fig 9. Dataset visualization diagram.**

## Experimental environment and parameter settings

The experimental platform used in this study is a Windows 10 system equipped with an NVIDIA GeForce RTX 3060 graphics card with 12G of VRAM. The experimental environment includes the PyTorch 1.8.0 framework, CUDA 11.0, and CUDNN 8.0.4 deep neural network acceleration libraries for model development and training.

During the training process, the setting of hyperparameters impacts the final performance and results. To ensure the rigor of the experiments, all models in this study were trained

under the same environment and with identical hyperparameter configurations, and the random seed was set to 42 to guarantee the reproducibility and stability of the experiments. The training parameters are as follows: a learning rate of 0.01 , using Stochastic Gradient Descent (SGD) for parameter optimization. To control model complexity and prevent over-fitting, a weight decay coefficient of $5 \times 10 - 5$ was set. The number of training epochs was set to 200 , and the network loss function is illustrated in Fig 10. The loss function curve shows that the loss value rapidly decreases in the first 50 epochs and gradually stabilizes after 200 epochs, eventually reaching saturation. Despite minor fluctuations in the loss value, the predictions align well with the actual values. Hence, the output model after 200 training epochs is determined as the final chimney smoke detection model.

## Evaluation metrics

The evaluation metrics for object detection algorithms include detection accuracy and model complexity. Therefore, to comprehensively and objectively assess the performance of the improved YOLOv8n model, this study compared metrics such as Precision (P), Recall (R), F1 Score (F1), Mean Average Precision (mAP), and Detection Rate (FPS). The specific calculation formulas are as shown in Eq (13):

$$P = \frac{TP}{TP + FP}, R = \frac{TP}{TP + FN}, F1 = \frac{2 * P * R}{P + R} \tag{13}$$

Where TP, FP, and FN represent true positives, false positives, and false negatives, respectively.

The Mean Average Precision (mAP) is the average of the AP values for all categories, and it is commonly used to evaluate the overall performance of an object detection model. In this paper, mAP not only takes into account the detection precision of each category but also combines different Intersection over Union (IoU) thresholds to comprehensively measure the model's performance. In this study, we adopted the AP, AP50, and AP75 metrics commonly used in the COCO dataset to evaluate the precision of the model [37,38].

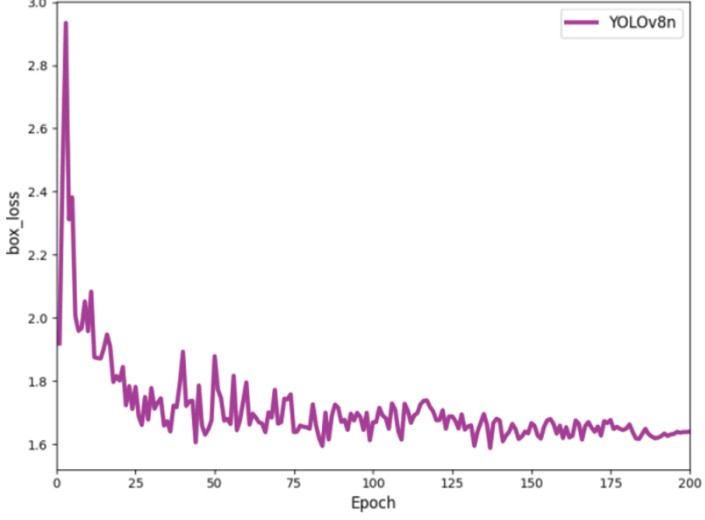
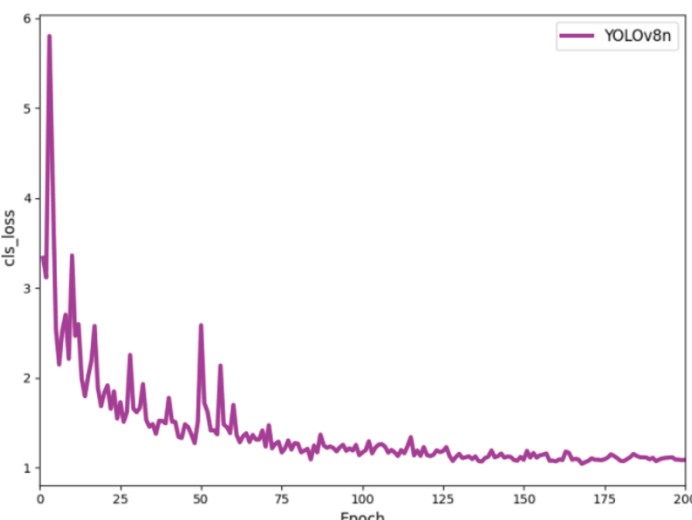

**Fig 10. Network loss function change diagram.**

AP: The Average Precision (AP) is calculated by computing the precision - recall curve based on different IoU thresholds, and then obtaining the area under this curve. Usually, multiple IoU values are considered comprehensively.

AP50: When the IoU threshold is set to 0.5, the calculated average precision mainly reflects the model's precision performance under a relatively low overlap degree.

AP75: When the IoU threshold is set to 0.75, the calculated average precision is often used to evaluate the model's precision under a relatively high overlap - degree requirement.

The calculation formulas for Average Precision (AP) and Mean Average Precision (mAP) are as shown in Eq (14):

$$AP = \sum_{i=1}^{n} (R_{i+1} - R_i) \max_{\tilde{R}:R \geq R_{i+1}} P(\tilde{R}), mAP = \frac{1}{N} \sum_{i=1}^{N} AP_i \tag{14}$$

Where i is the index value, P and R represent precision and recall, respectively, N is the number of categories, and $AP_i$ is the average precision for the ith category.

Detection rate can be quantified as frames per second (FPS), calculated as shown in Eq (15):

$$FPS = \frac{1}{t} \tag{15}$$

In the context of the study, $t$ represents the average time taken for detecting each image. Using the aforementioned formula, the number of images processed per second can be calculated, which is a key metric for assessing the model's real-time performance in image processing.

Additionally, the model's complexity is evaluated by quantifying the number of floating-point operations per second (FLOPs) and the number of parameters (Parameters). FLOPs serve as an indicator of the computational workload of the model [38], while Parameters refer to the total number of weights and biases that need to be learned in the neural network model. The size of the parameters directly influences the model's storage requirements and computational overhead [39].

## Experiments and analysis

**Loss function ablation experiment.** The WIoU utilized in this study has three versions: v1, v2, and v3. WIoUv1 established an attention-based bounding box loss, whereas WIoUv2 and v3 added a focusing mechanism through the construction of a gradient gain calculation method. v2 employs a monotonic focusing mechanism, while v3 utilizes a non-monotonic focusing mechanism. By optimizing YOLOv8n with these three different versions of WIoU loss functions, the study aimed to identify the most suitable loss function for its dataset. The experimental results are presented in the following Table 1.

According to Table 1, the application of the WIoU loss function has virtually no effect on model size or inference speed. Upon adopting the WIoUv2 loss function, all metrics exhibit

**Table 1. Performance comparison of three loss functions.**

| Model | P / % | R / % | F 1 / % | AP / % | AP 50 / % | AP 75 / % | mAP 0.5 / % | FPS(f/s) | FLOPs/G |
|---|---|---|---|---|---|---|---|---|---|
| YOLOv8n(CIoU) | 90.3 | 89.3 | 88.6 | 40.9 | 90.1 | 59.8 | 90.1 | 45.5 | 8.9 |
| YOLOv8n-WIoUv1 | 91.8 | 92.1 | 89.7 | 42.1 | 91.4 | 63.1 | 91.4 | **45.5** | 8.9 |
| YOLOv8n-WIoUv2 | **92.5** | **92.7** | **90.5** | 42.7 | 92.1 | 63.8 | **92.1** | 45.3 | **8.9** |
| YOLOv8n-WIoUv3 | 92.1 | 92.3 | 90.2 | 42.3 | 92.0 | 63.3 | 92.0 | 45.1 | 8.9 |

optimal performance: AP increases to 42.7% (a 1.8% improvement compared to CIoU), AP50 achieves 92.1% (+2.0%), AP75 rises to 63.8% (+4.0%), and mAP0.5 also improves to 92.1% (+2.0%). The monotonic focusing mechanism introduced by WIoUv2 enhances model performance through prioritizing critical regions during training, accelerating convergence, and alleviating gradient vanishing issues. This mechanism is especially beneficial for deeper networks as it efficiently guides the model to focus on crucial areas while maintaining training stability and efficiency. Consequently, this study utilizes WIoUv2 along with its monotonic focusing mechanism to optimize both bounding box regression and classification losses, thereby better adapting to deeper network architectures and ultimately enhancing model performance and training efficiency.

Visual analysis of mAP during training, illustrated in Fig 11, shows YOLOv8n optimized with WIoUv2 converging by epoch 25, achieving faster convergence and sustaining the highest mAP0.5 curve against YOLOv8n-CIoU, YOLOv8n-WIoUv1, and YOLOv8n-WIoUv2 variants. This underscores the suitability of WIoUv2 with a monotonic focusing mechanism for the dataset in this study, boosting model performance and speeding up convergence.

**Ablation experiment on optimization modules.**  To determine the impact of each optimization module on network performance, ablation experiments were conducted by adding different modules to the YOLOv8n model. Specifically, three groups of experiments were designed to analyze various improvements, with each group using the same training parameters but testing on different models. The results of these tests are presented in Table 2. In the

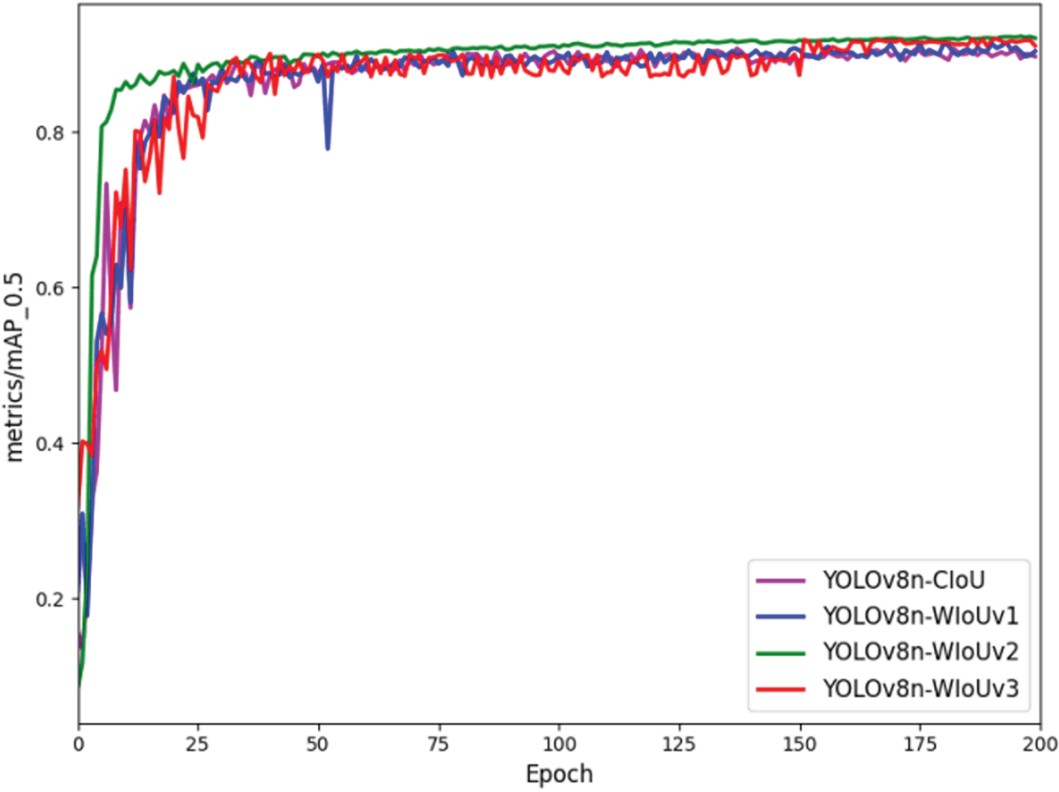

**Fig 11. mAP0.5 Curves of various models.**

**Table 2. Performance comparison of ablation experiments on optimization modules.**

| Model | BC2f | WIoUv2 | GSConv | P% | R/% | F1/% | AP / % | AP 50 / % | AP 75 / % | mAP0.5/% | FPS(f/s) | FLOPs/G | Parameters/M |
|---|---|---|---|---|---|---|---|---|---|---|---|---|---|
| **YOLOv8n** | × | x | x | 90.3 | 89.3 | 88.6 | 40.9 | 90.1 | 59.8 | 90.1 | 45.5 | 8.9 | 3.15 |
| **YOLOv8n-B** | √ | × | × | 92.1 | 93.7 | 91.1 | 42.2 | 93.4 | 64.1 | 93.4 | 41.3 | 9.2 | 3.51 |
| **YOLOv8n-BW** | √ | √ | x | 93.0 | 94.2 | 92.5 | 42.6 | 64.7 | 41.1 | 94.1 | 41.1 | 9.2 | 3.51 |
| **YOLOv8n-BWG** | √ | √ | √ | **93.5** | **94.6** | **92.8** | **43.7** | **94.3** | **64.9** | **94.3** | **55.2** | **6.3** | **2.32** |

table, a "√" indicates a strategy used in the improved model, while an "×" denotes a strategy not utilized in the improved model.

An analysis of Table 2 reveals the following: In the YOLOv8n-B model, the BC2f structure designed in this paper is integrated into the backbone network of YOLOv8n. As a result, its F1 score increases by 2.5%, mAP rises by 3.3%, AP improves by 1.3%, AP50 increases by 3.3%, and AP75 goes up by 4.3%. However, the detection speed decreases by 4.2 f/s, the floating-point operations of the model increase by 0.3 G, and the number of parameters (Parameters) increases by 0.36 M. This indicates that integrating the BC2f structure can enhance the network's focus on chimney smoke, effectively suppress invalid features and noise, improve the model's mean average precision, and also enhance the target localization accuracy at a higher IoU threshold (AP75). Meanwhile, it will increase the computational burden and the model size.

The YOLOv8n-BW model upgrades the original CIoU loss function with the WIoUv2 function on the basis of the YOLOv8n-B model. It has little impact on the detection speed and model complexity. Instead, the F1 score increases by 1.5% and the mAP rises by 1.3%. In terms of AP, AP improves by 0.4%, AP50 increases by 0.7%, and AP75 goes up by 0.6%. This shows that the WIoUv2 loss function can effectively improve the model's precision at different IoU thresholds by enhancing the attention to small-sample targets, especially excelling in detecting small targets, while having very little influence on speed and complexity.

The YOLOv8n-BWG model introduces the lightweight convolution GSConv to replace the standard convolution on the basis of the YOLOv8n-BW model. It can be observed that the F1 score increases by 0.3% and the mAP rises by 0.2%. In terms of AP, AP improves by 1.1%, AP50 increases by 0.2%, and AP75 goes up by 0.2%. The detection speed increases by 14.1 f/s, the floating-point operations of the model are reduced by 2.9 G, and the number of parameters is decreased by 1.19 M. This indicates that adopting the GSConv convolution can not only ensure the model's precision but also reduce the number of parameters in the model and improve the detection speed.

Precision (P), Recall (R), and Mean Average Precision (mAP) curves, depicted in Fig 12, visually represent the impact of various modules on model performance. The proposed model consistently outshines the baseline across these metrics, with the YOLOv8n-BWG model achieving the highest precision of 93.5% after 50 epochs, as shown in Fig 12(a). This model also demonstrates quicker convergence, around 25 epochs, compared to other models' 50 epochs, as illustrated in Fig 12(b). The mAP0.5 curve of the YOLOv8n-BWG model remains superior, reaching 94.3% after 50 epochs, underscoring its enhanced performance in chimney smoke detection, as seen in Fig 12(c).

In conclusion, the YOLOv8n-BWG model proposed in this study not only converges the fastest but also performs best across all performance indicators.

**Comparison experiment of different models.** To validate the performance of the improved model, this study selected the current mainstream two-stage detection algorithm Faster R-CNN and single-stage detection algorithms SSD, YOLOv4-Tiny, YOLOv5s,

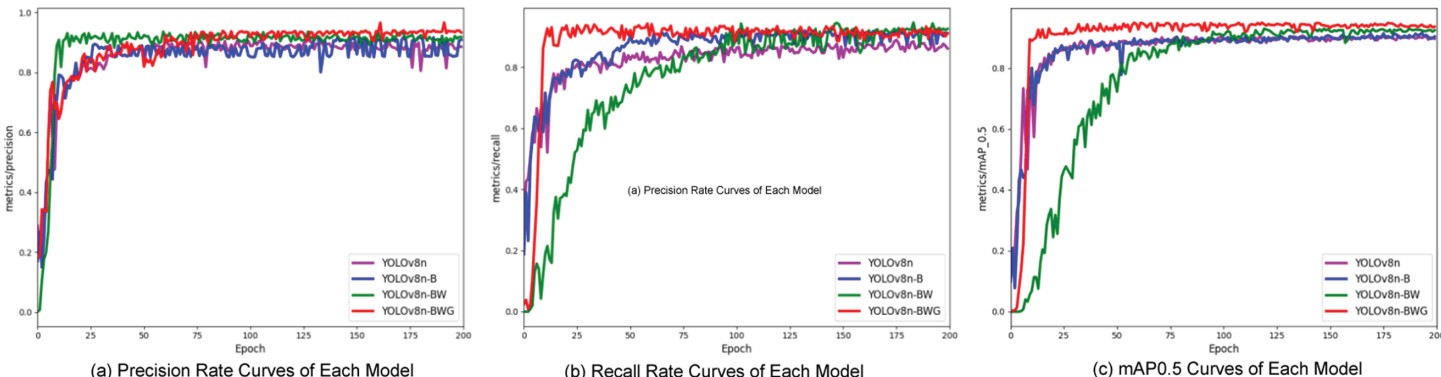

**Fig 12. Comparative diagrams of different indicators.**

YOLOv7-Tiny, YOLOv10n, and YOLOv11n for comparative experiments. Each model used the same hyperparameters, and the results are resented in Table 3.

Among all models, the floating-point operations (FLOPs) of Faster R-CNN stand at 135G, with a parameter size of 69.7M. The FLOPs and parameter size of the Faster R-CNN model are too large to be embedded in drones for use. The single-stage detection algorithm SSD model has lower accuracy and relatively high FLOPs and parameters, making it unsuitable for meeting the requirements of drone-based chimney smoke detection. The YOLOv4-Tiny and YOLOv5s models have FLOPs of 21.7G and 16.4M, respectively, and parameter sizes of 10.4G and 7.2M. This indicates that the FLOPs and parameter sizes of the YOLOv4-Tiny and YOLOv5s models are too large to meet the requirements for drone deployment. The YOLOv7-Tiny model has higher accuracy but slower detection speed, which is not conducive to mobile deployment on drones.

Although the YOLOv10n and YOLOv11n models have relatively high accuracy, they are slightly inferior in accuracy compared to the proposed YOLOv8n-BGW model, which performs the best among all models, achieving an accuracy of 94.3%. Additionally, the YOLOv8n-BGW model excels in computational performance, with the lowest FLOPs and parameter size of only 6.3G and 2.32M, respectively. This shows that the YOLOv8n-BWG model meets the requirements for both accuracy and complexity in chimney smoke detection. Moreover, the YOLOv8n-BGW model achieves 55.2 frames per second (FPS), meeting the requirements for mobile deployment on drones. In summary, the proposed YOLOv8n-BGW model exhibits the best performance among all models, can be effectively embedded in drones, and meets the accuracy requirements for detection.

**Table 3. Performance comparison of different models.**

| Model | P/% | R/% | F1/% | mAP0.5/% | FPS(f/s) | FLOPs/G | Parameter/M |
|---|---|---|---|---|---|---|---|
| **FASTER RCNN** | 82.3 | 85.7 | 80.1 | 83.1 | 2.1 | 135 | 69.7 |
| **SSD** | 78.2 | 79.3 | 76.3 | 79.1 | 28 | 29.2 | 18.7 |
| **YOLOV4-TINY** | 85.7 | 84.2 | 82.1 | 85.5 | 30.3 | 21.7 | 10.4 |
| **YOLOV5S** | 90.1 | 87.5 | 85.6 | 89.3 | 35.6 | 16.4 | 7.2 |
| **YOLOV7-TINY** | 87.3 | 88.5 | 86.2 | 88.5 | 39.1 | 9.8 | 6.2 |
| **YOLOv10n** | 90.1 | 81.4 | 88.4 | 90.6 | 47.3 | 6.9 | 3.18 |
| **YOLOv11n** | 91.3 | 92 | 89.9 | 91.2 | 49.5 | 6.7 | 2.69 |
| **YOLOv8n-BWG** | **93.5** | **94.6** | **92.8** | **94.3** | **55.2** | **6.3** | **2.32** |

To intuitively compare the comprehensive performance of the models, this paper evaluates eight models across multiple metrics, including precision (P), recall (R), F1 score, mAP0.5, average inference time, floating-point operations, and number of parameters. The results are visualized using a radar chart, as shown in Fig 13. Each curve in the radar chart corresponds to one model. The closer the intersection point of the curve with the axis is to the outer edge, the better the model performs in that metric. The larger the area enclosed by the curve, the more superior the overall performance of the model.

As can be seen from Fig 13, although some models (such as YOLOv10n and YOLOv11n) also show good performance in terms of lightweight design, the YOLOv8n-BGW model stands out more prominently in comprehensive performance, especially in model accuracy. This advantage makes the YOLOv8n-BGW more competitive in practical applications and better able to meet the demands of high-precision detection.

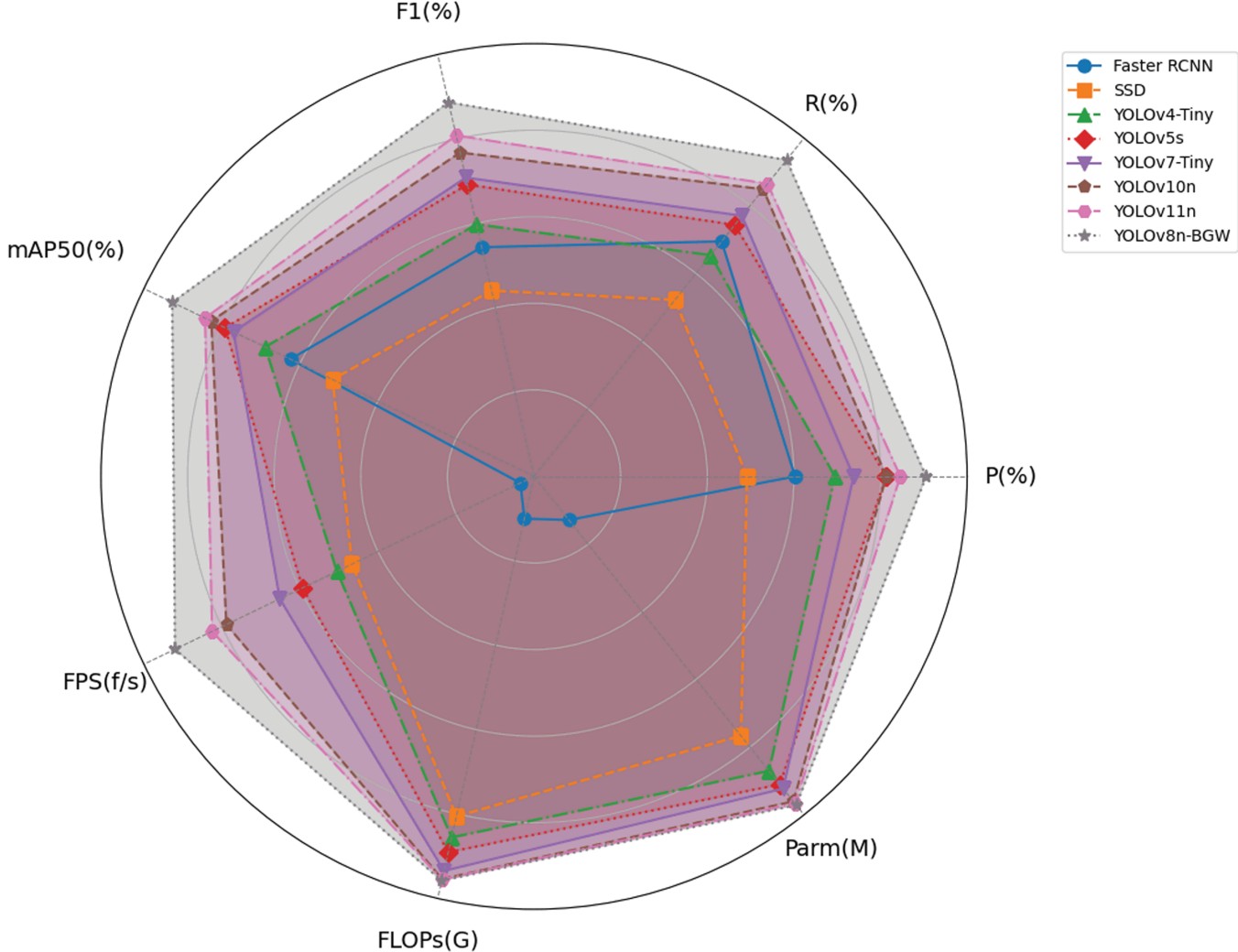

**Fig 13. Radar chart of multi-model performance comparison.**

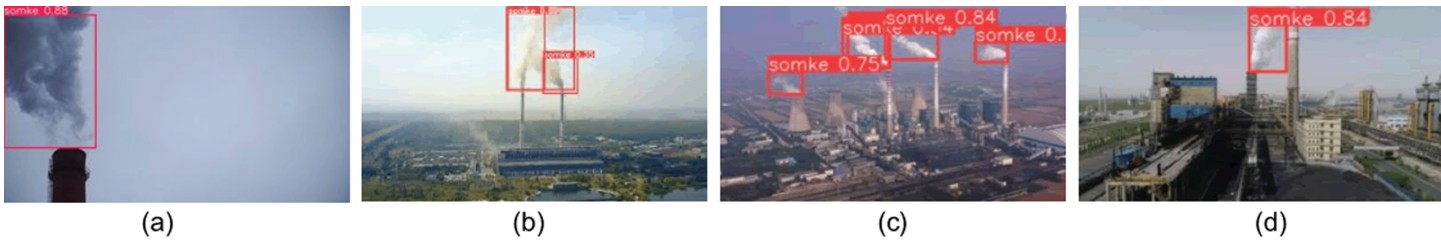

**Fig 14. YOLOv8n detection results diagram.**

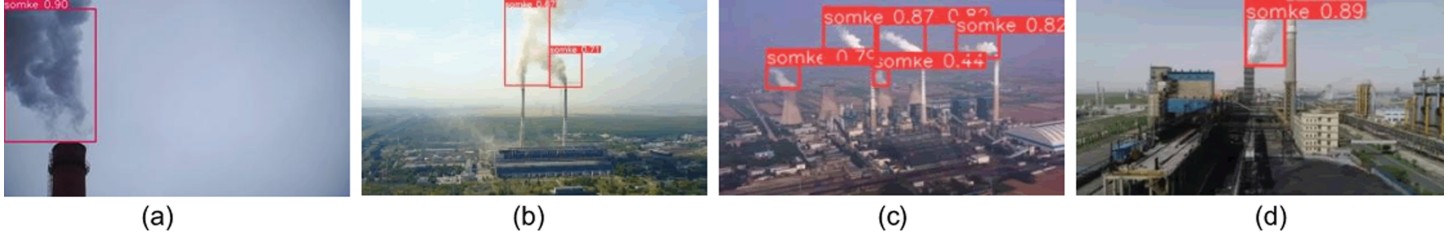

**Fig 15. YOLOv8n-BWG detection results diagram.**

## Detection results comparison

To observe the actual detection effects, the improved model YOLOv8n-BWG from this study was tested against the YOLOv8n on a test set. The detection results of YOLOv8n are shown in Fig 14, while the results of YOLOv8n-BWG are presented in Fig 15.

Upon comparing Figs 15(a) and 14(a), we observe enhanced detection accuracy in the YOLOv8n-BWG model over the YOLOv8n. Analysis of Figs 15(b) and 14(b) highlights the YOLOv8n-BWG model's superior precision in smoke localization. The comparison between Figs 15(c) and 14(c) indicates that the YOLOv8n-BWG model adeptly identifies small targets without omissions, unlike the YOLOv8n model. Additionally, Figs 15(d) and 14(d) underscore the YOLOv8n-BWG model's improved detection accuracy over YOLOv8n. The YOLOv8n-BWG model, therefore, markedly enhances detection accuracy and the detection of smaller smoke targets at extended distances, proving to be more effective in practical UAV detection scenarios.

## Applications

Experimental results affirm the YOLOv8n-BWG model's efficacy in UAV applications, satisfying both accuracy and complexity requisites for chimney smoke detection. Standard UAVs, however, face challenges in pinpointing the precise location of smoke due to their reliance on 2D image-based positional information [40].

To mitigate this, the integration of a binocular stereo vision system is proposed. This system leverages depth information from dual-camera images to compute the smoke's 3D location. With this enhancement, UAVs equipped with the YOLOv8n-BWG model and a stereo vision system can identify chimney smoke and calculate its 3D position, as depicted in Fig 16. The process involves initial smoke area detection, followed by 3D positioning with the stereo vision system, and culminates in the transmission of both 2D and 3D smoke area data for backend analysis. This approach enables UAVs to localize smoke sources with greater accuracy, offering more detailed and reliable data for environmental monitoring.

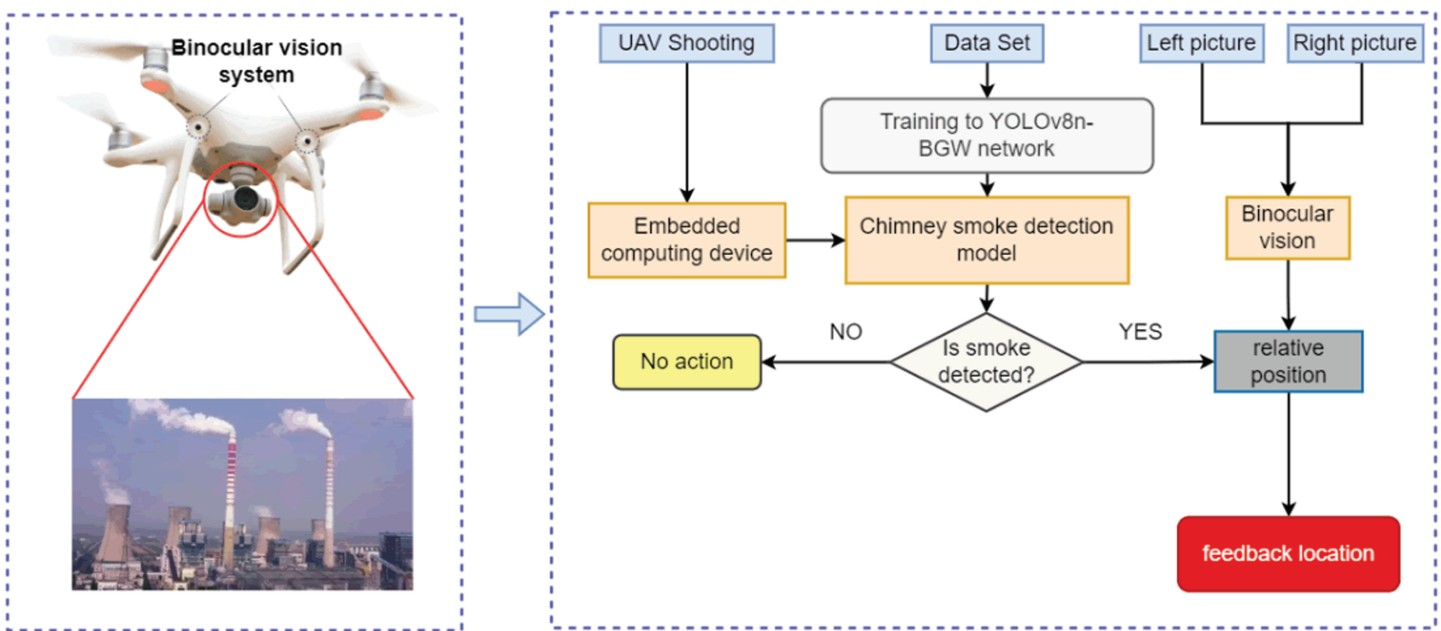

**Fig 16. UAV smoke monitoring and precise traceability process diagram.**

The binocular stereo vision system, comprising left and right cameras, employs ranging technology to ascertain the 3D position of objects. This method involves measuring the disparity between images captured by each camera to determine the object's distance from the cameras. The geometry of this vision system is illustrated in Fig 17, where OL and OR represent the apertures' centers of the left and right cameras, respectively [41]. For a target point P in space, its projections PL and PR in the left and right cameras allow for distance calculation using the principle of similar triangles, as demonstrated in Eq (16).

$$\frac{Z}{Z-f} = \frac{b}{b - u_L - u_R} \tag{16}$$

Where $f$ represents the focal length, b denotes the baseline distance, and $z$ is the distance between the measured point and the camera. The formula can be simplified as shown in Eq (17):

$$Z = \frac{fb}{d}, (d = u_L + u_R) \tag{17}$$

Where $d$ is the disparity, representing the directional difference caused by observing the same point from two different points.

To obtain the three-dimensional coordinates of point P, it is necessary to calculate its coordinates on the X and Y axes of the image plane. Assuming the coordinates of point P on the image plane are XP and YP, and its coordinates in pixel space are $u$ and $v$. The conversion from camera coordinates to image plane coordinates can be achieved through similar triangles in the imaging model, as expressed in Eq (18).

$$\frac{X}{X_P} = \frac{Y}{Y_P} = \frac{Z}{f} \tag{18}$$

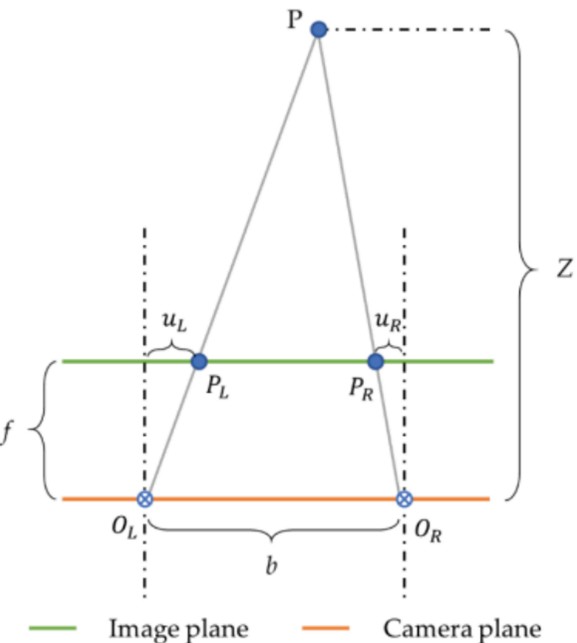

**Fig 17. Geometric model of the binocular system.**

Similarly, converting image plane coordinates to pixel coordinates requires scaling and translation operations, with the specific calculation formula shown in Eq (19).

$$\begin{cases} u = \alpha_x X_P + c_x \\ v = \alpha_y Y_p + c_y \end{cases} \tag{19}$$

Where $\alpha$ and $c$ represent the scaling factor and translation factor, respectively, having different values on each axis. By jointly using Eqs (18) and (19), Eq (20) can be derived, representing the camera coordinates of point P.

$$\begin{cases} X = \frac{Z(u-c_x)}{f_x} \\ Y = \frac{Z(v-c_y)}{f_y} \end{cases} \tag{20}$$

Where $f_x = \alpha_x f$ and $f_y = \alpha_y f$, and they are measured in pixels. Typically, $f_x, f_y, c_x$, and $c_y$ are referred to as the intrinsic values of the camera. Through these formula calculations, the spatial coordinates $(x,y,z)$ of the smoke can be real-time feedback.

## Conclusion

Effective smoke monitoring can effectively manage and reduce the adverse impacts of industrial emissions on the environment and human health, providing substantial support for air quality improvement and sustainable development. To consider embedding the model into UAV devices, based on YOLOv8n, the BC2f structure, WIoUv2 function, and GSConv convolution were integrated into the YOLOv8n network, resulting in the improved model called YOLOv8n-BWG.

In this study, the BC2f structure effectively reduced computational and storage consumption through an adaptive query mechanism while enhancing the model's perception of input content, focusing on smoke targets, suppressing ineffective features and noise, thereby reducing false detection rates. The WIoUv2 function and dynamic sample allocation strategy re-optimized the loss function, significantly improving the model's focus on small targets and reducing the miss rate of small target smoke. The lightweight GSConv convolution reduced the number of parameters while maintaining high accuracy, effectively lowering model complexity while ensuring computational efficiency.

The results show that on a custom dataset, compared to the YOLOv8n model, the YOLOv8n-BWG model increased the average recognition precision mAP by 4.2%, improved recognition speed by 21.3% per second, reduced FLOPs by 28.9%, and reduced the number of parameters by 26.3%. Through final detection results, it was found that the YOLOv8n-BWG model had higher detection accuracy, faster detection speed, and effectively reduced the miss rate of small targets compared to the YOLOv8n model. This paper also uses a binocular stereo vision system to calculate the exact location of smoke areas, providing more comprehensive and reliable environmental sensing data.

Currently, the YOLOv8n-BWG model achieves a detection speed of 55.2 frames on computer devices and only 13.2 frames on UAV edge devices. Therefore, there is room for optimizing detection speed on edge devices. Future plans include accelerating the model and improving detection speed through model pruning [42] and deploying TensorRT [43].

## Acknowledgments

The authors extend their appreciation to the Key Scientific Research Project of Higher Education Institutions in Henan Province (Grant No. 24B520035) and the Scientific and Technological Breakthrough Project of Henan Province (Grant No. 252102111173).

## Author contributions

**Conceptualization:** Yaojun Zhang.

**Data curation:** Guiling Wu.

**Formal analysis:** Yaojun Zhang.

**Funding acquisition:** Yaojun Zhang.

**Investigation:** Guiling Wu.

**Methodology:** Yaojun Zhang, Guiling Wu.

**Project administration:** Yaojun Zhang.

**Validation:** Guiling Wu.

**Visualization:** Yaojun Zhang.

**Writing – original draft:** Yaojun Zhang.

**Writing – review & editing:** Guiling Wu.

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
