## [Decision Letter · Decision Letter 0]

28 Jan 2025

PONE-D-25-00562YOLOv8n-BWG-Enhanced Drone Smoke Detection: Advancing Environmental Monitoring EfficiencyPLOS ONE

Dear Dr. Wu,

Thank you for submitting your manuscript to PLOS ONE. After careful consideration, we feel that it has merit but does not fully meet PLOS ONE’s publication criteria as it currently stands. Therefore, we invite you to submit a revised version of the manuscript that addresses the points raised during the review process.

Please submit your revised manuscript by Mar 14 2025 11:59PM. If you will need more time than this to complete your revisions, please reply to this message or contact the journal office at plosone@plos.org. Please include the following items when submitting your revised manuscript:

We look forward to receiving your revised manuscript.

Kind regards,

Yile Chen, Ph.D. in Architecture

Academic Editor

PLOS ONE

Journal Requirements:

2. In your Methods section, please include additional information about your dataset and ensure that you have included a statement specifying whether the collection and analysis method complied with the terms and conditions for the source of the data.

“This research was funded by the Key Scientific Research Project of HigherEducation Institutions in Henan Province (Grant No. 24B520035) and the Scientific andTechnological Breakthrough Project of Henan Province (Grant No. 242102210090).”

Reviewers' comments:

Reviewer's Responses to Questions

**Comments to the Author**

1. Is the manuscript technically sound, and do the data support the conclusions?

Reviewer #1: Yes

Reviewer #2: Yes

2. Has the statistical analysis been performed appropriately and rigorously? 

Reviewer #1: Yes

Reviewer #2: Yes

3. Have the authors made all data underlying the findings in their manuscript fully available?

Reviewer #1: No

Reviewer #2: Yes

4. Is the manuscript presented in an intelligible fashion and written in standard English?

Reviewer #1: Yes

Reviewer #2: Yes

5. Review Comments to the Author

Reviewer #1: 1. Although the process of collecting, labelling, and enhancing the dataset is described, the division of the dataset in the experiment is not clearly stated. It is suggested to add the division ratio of the dataset.

2. The experimental environment and parameter settings provide a detailed description of the hyperparameters required for training, but there is no clear description of the parameter tuning process. Please provide a description of how the optimal parameters were selected. In addition, measures to ensure model stability (e.g., setting random seeds) are missing from the experimental description.

3. Parameters can be used to measure the lightweight performance of a model, but there is a lack of descriptions and comparisons of Parameters in the comparison experiments of different models in Table 3. Please add the Parameters metrics in Table 3 and add relevant descriptions in the text.

4. The conclusion is a bit too simple, and it is suggested to add what problems are solved by introducing BC2f and other modules.

5. There are some awkward sentences in the text, such as ‘The single-stage SSD model, despite lower accuracy, does not meet chimney smoke’ should be The single-stage SSD model does not meet the requirements for chimney exhaust due to its low accuracy.’ Please check the grammar of the manuscript for clarity.

Reviewer #2: (1)Why not use higher version of YOLO, such as YOLOv10?

(2)Some paragraphs are too long and difficult to follow, e.g. Lines 39~67. Please divide them into several short paragraphs to improve the readability.

(3)The review of the related works and comparison experiments can be more sufficient. Please carefully read and compare (if applicable) the following papers on SOTA detection. [DOI: 10.1016/j.patcog.2023.109878; 10.3390/s20041010] If the authors cannot employ these methods or compare their method with these methods, at least they could introduce/mention these technologies in related sections to improve the quality of the survey.

(4)Please provide and label the reference indices of the compared methods in the figures and tables, and then the readers can judge whether the compared methods are SOTA.

(5)Please use bold font to label the best results in all tables.

(6)In Figs. 13(a) and 14(a), are three some overlap problems? One smoke or two smokes?

6. PLOS authors have the option to publish the peer review history of their article (what does this mean?). If published, this will include your full peer review and any attached files.

Reviewer #1: No

Reviewer #2: No

---

## [Author Response · Author response to Decision Letter 1]

4 Mar 2025

Thank you for your valuable comments and suggestions on our manuscript. We have uploaded the Response to Reviewers, Revised Manuscript with Track Changes, and Manuscript as required. We sincerely appreciate your time and effort in reviewing our work.

---

## [Decision Letter · Decision Letter 1]

24 Mar 2025

YOLOv8n-BWG-Enhanced Drone Smoke Detection: Advancing Environmental Monitoring Efficiency

PONE-D-25-00562R1

Dear Dr. Wu,

We’re pleased to inform you that your manuscript has been judged scientifically suitable for publication and will be formally accepted for publication once it meets all outstanding technical requirements.

Kind regards,

Yile Chen, Ph.D. in Architecture

Academic Editor

PLOS ONE

Additional Editor Comments (optional):

Reviewers' comments:

Reviewer's Responses to Questions

**Comments to the Author**

1. If the authors have adequately addressed your comments raised in a previous round of review and you feel that this manuscript is now acceptable for publication, you may indicate that here to bypass the “Comments to the Author” section, enter your conflict of interest statement in the “Confidential to Editor” section, and submit your "Accept" recommendation.

Reviewer #1: All comments have been addressed

Reviewer #2: All comments have been addressed

2. Is the manuscript technically sound, and do the data support the conclusions?

Reviewer #1: Yes

Reviewer #2: Yes

3. Has the statistical analysis been performed appropriately and rigorously? 

Reviewer #1: Yes

Reviewer #2: Yes

4. Have the authors made all data underlying the findings in their manuscript fully available?

Reviewer #1: (No Response)

Reviewer #2: Yes

5. Is the manuscript presented in an intelligible fashion and written in standard English?

Reviewer #1: (No Response)

Reviewer #2: Yes

6. Review Comments to the Author

Reviewer #1: (No Response)

Reviewer #2: Effective smoke monitoring can effectively manage and reduce the adverse impacts of industrial emissions on the environment and human health, providing substantial support for air quality improvement and sustainable development. To consider embedding the model into UAV devices, This paper integrated the BC2f structure, WIoUv2 function, and GSConv convolution into the YOLOv8n network, resulting in the improved model called YOLOv8n-BWG.

7. PLOS authors have the option to publish the peer review history of their article (what does this mean?). If published, this will include your full peer review and any attached files.

Reviewer #1: No

Reviewer #2: No

---

## [Editor Report · Acceptance letter]

PONE-D-25-00562R1

PLOS ONE

Dear Dr. Wu,

I'm pleased to inform you that your manuscript has been deemed suitable for publication in PLOS ONE. Congratulations! Your manuscript is now being handed over to our production team.

Kind regards,

on behalf of

Dr. Yile Chen

Academic Editor

PLOS ONE